# Relative Pose Estimation between Image Object and ShapeNet CAD Model for Automatic 4-DoF Annotation

Soon-Yong Park [1,*] , Chang-Min Son [2] , Won-Jae Jeong [2] and Sieun Park [2]

[1] School of Electronics Engineering, Kyungpook National University, Daegu 41566, Republic of Korea
[2] Graduate School of Electronic and Electrical Engineering, Kyungpook National University, Daegu 41566, Republic of Korea
* Correspondence: sypark@knu.ac.kr; Tel.: +82-53-950-7575

**Abstract:** Estimating the three-dimensional (3D) pose of real objects using only a single RGB image is an interesting and difficult topic. This study proposes a new pipeline to estimate and represent the pose of an object in an RGB image only with the 4-DoF annotation to a matching CAD model. The proposed method retrieves CAD candidates from the ShapeNet dataset and utilizes the pose-constrained 2D renderings of the candidates to find the best matching CAD model. The pose estimation pipeline consists of several steps of learned networks followed by image similarity measurements. First, from a single RGB image, the category and the object region are determined and segmented. Second, the 3-DoF rotational pose of the object is estimated by a learned pose-contrast network only using the segmented object region. Thus, 2D rendering images of CAD candidates are generated based on the rotational pose result. Finally, an image similarity measurement is performed to find the best matching CAD model and to determine the 1-DoF focal length of the camera to align the model with the object. Conventional pose estimation methods employ the 9-DoF pose parameters due to the unknown scale of both image object and CAD model. However, this study shows that only 4-DoF annotation parameters between real object and CAD model is enough to facilitates the projection of the CAD model to the RGB space for image-graphic applications such as Extended Reality. In the experiments, performance of the proposed method is analyzed by using ground truth and comparing with a triplet-loss learning method.

**Keywords:** pose estimation; CAD retrieval; ShapeNet; image similarity; 4-DoF annotation; extended reality

## 1. Introduction

Estimating the three-dimensional (3D) pose of an object in an image captured with a camera is an important problem in computer and robot vision. Estimating the 3D pose of an object refers to determining the 3D rotation and translation—that is, six degrees of freedom (6-DoF)—of the object from a certain reference coordinate system to the camera coordinate system. Here, the 3D rotation and translation refer to the 3D rigid transformation from the reference coordinate system to the camera coordinate system. Moreover, in case that 3D CAD model retrieval from the reference coordinate system is considered at the same time, the 3D pose information must be represented in 9-DoF due to the unknown scale between the object and the corresponding CAD model.

Estimating the 3D pose of an object in an image captured with a camera has several application areas. For example, in order for a robot to pick up an object, it has to estimate the 3D pose of the object to determine the picking points of its own hand. Another application area is autonomous driving vehicles. For example, images of road conditions and circumstances are captured with a camera installed on an autonomous driving vehicle. These images are then used to estimate the 3D pose of other vehicles on the road to predict their motion.

The technology for estimating the 3D pose of an object can also be used in extended reality (XR). As shown in Figure 1, if the 3D pose of the bench object shown in the camera

image is estimated, the estimated rotation and translation information is used to transform the 3D CAD model of the bench into the camera coordinate system and project it onto the image to align it perfectly with the real image. XR techniques can be applied to remove the bench from the image or to change the shape of the bench bench using this alignment result. The 3D pose of the bench needs to be precisely estimated to align the chair in the image with the CAD model, as shown in Figure 1. In this case, the CAD model is generally represented with an unknown scale. Thus, 3D pose between the CAD model and the real bench object must be represented in 9-DoF parameters.

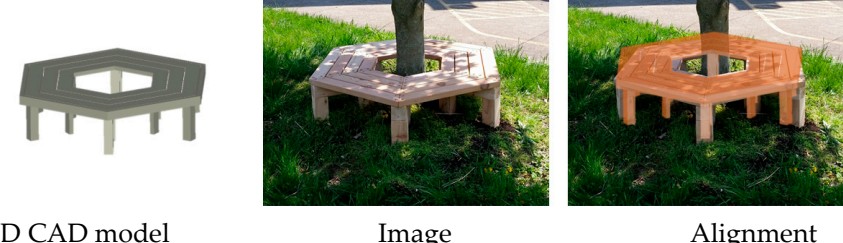

3D CAD model              Image              Alignment

**Figure 1.** 3D pose annotation of a real object can be used to project and align the matching CAD model to the object region in an image.

Various approaches have been proposed for estimating the 3D pose of an object using images. Most previous studies on 3D pose estimation employed two types of techniques. The first technique is to utilize the 3D depth information of the object in an image using a depth camera. The second technique is to perform 3D-to-3D matching between the CAD model and the 3D shape information of the object extracted with a stereo matching technique using two or more images [1–3]. 3D depth or shape of the real object is very useful information for 3D pose estimation. However, acquiring the 3D information of an object from the image is difficult if only a single RGB image is used. In this study, we address a 3D pose estimation using only a single RGB image.

In this paper, we propose a new framework to estimate the relative 3D pose between an object in a single RGB image and its CAD correspondence in an Internet dataset. The 3D pose information is represented by only 4-DoF parameters between the object and the model. This also provides the 4-DoF annotation to the CAD model to align the projection of the CAD model to the object region in the RGB image. The proposed framework utilizes a method that retrieves from the CAD dataset a 3D mesh model with the same class as the object detected in the image.

Figure 2 shows the proposed pose estimation framework, and the overall process is as follows. First, the object is detected from an RGB input image using a deep learning network to determine its class and bounding box (BB). Second, the 3-DoF rotational pose of the object—azimuth, elevation, and inplane rotation—is determined using only the BB region of the input image. This process uses a pretrained deep learning network for pose estimation. Third, candidate 3D CAD models with the same class are retrieved from the CAD database using the 3D rotational pose and class information of the object. Here, the estimated rotational pose is used to rotate the CAD model, and two-dimensional (2D) rendered images of the CAD models are generated through rendering. Fourth, the similarity between the rendered images of the candidate CAD models and the actual object image is measured. Among the candidate CAD models, the model that best matches the actual object is selected. Finally, the 3D mesh model of the selected CAD model is placed in a fixed distance on the $z$-axis from the camera and projected onto the actual image by changing the focal length. This process is repeated, and the 1-DoF focal length is determined by selecting the value that produces the best alignment with the object region of the actual image.

The contributions of this study are summarized as follows:

(1)    The 4-DoF pose estimation pipeline from a single RGB image is proposed.

(2)    The 4-DoF pose annotation database can be generated to align the CAD model of a real object in the image plane.

(3)    Three image similarity criteria are proposed to match deep features between rendered CAD and real object images.

Section 2 presents a review of various existing methods for estimating the 3D pose of an object detected in an image. Then, Section 3 presents the proposed framework in detail, and Section 4 describes the experiment for estimating the 3D pose of objects detected in real images collected by us. Finally, Section 5 presents the discussion.

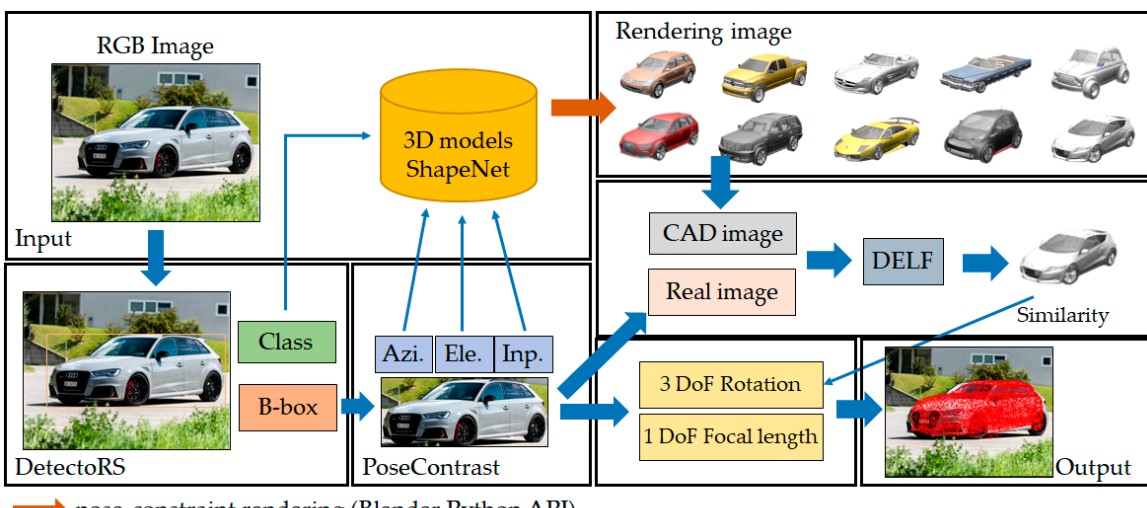

**Figure 2.** The proposed 4-DoF pose estimation pipeline.

## 2. Literature Review

### 2.1. Retrieval of CAD Models

With regard to one of the SHREC tracks, Hua B. et al. [4] introduced various techniques for retrieving CAD models and estimating the pose of an object using the 2D information and 3D depth information of the object from an RGB-D image. In their study, a 3D CAD model dataset called ObjectNN was provided to track participants, and the performances of various algorithms were compared. ObjectNN was composed of 1667 RGB-D images obtained from the SceneNN [5] dataset and 3308 CAD models of 20 categories obtained from ShapeNet [6]. Their study compared the performances of six deep-learning-based algorithms and three general feature-based algorithms, and the results of the experiment showed that the model retrieval performance of the deep-learning-based algorithms was superior to that of the feature-based algorithms.

### 2.2. Retrieval of CAD Models and Pose Estimation

Gümeli C. et al. [7] proposed a method most similar to the framework proposed in this paper for estimating the 3D pose of an object. Their study also used RGB images as the input and showed the process of estimating the pose of the CAD model with respect to the camera coordinate system and projecting it onto the image as the final result. In their study, the ResNet-50-FPN [8,9] was used to detect objects, and Mask-RCNN [10] was used to segment the region of the object. Furthermore, to utilize the 3D distance information of an object, the depth information was obtained from a single input image using MonoDepth deep learning convolutional neural network (CNN) based on multiscale feature fusion [11]. The normal vector image of the surface of the object was recalculated using the depth and segmentation information. In addition, the 3D shape and normal vector information of the CAD model and the 2D color image and 3D normal vector image of the object were used as the input to a deep learning network, and the pose of the CAD model with the best match was estimated using the deep learning network. Moreover, the ScanNet25K [12] image data

and CAD models provided by Scan2CAD [13] were used in their experiment. The error measurement of the method proposed in their paper utilized the 3D annotation information of Scan2CAD, and the accuracy of translation, rotation, and scale ratio was 27.4%.

In the previously described paper by Gümeli C. et al. [7], 3D CAD models of Scan2CAD [13] were used. In Scan2CAD, a deep learning network was proposed for detecting an object in an RGB-D input image and aligning a CAD model matching the detected object with the RGB-D image. In addition, 3D CAD models, 3D scan data obtained with a depth camera, and 3D pose annotation data of each model between the CAD model and scan data were provided to train the network. For the 3D scan data, 1506 data points imported from ScanNet [12] were used, and the annotation information of keypoints between 14,225 ShapeNet CAD models and scan data were provided. In their study, a 3D CNN-based deep learning network was proposed to align the 3D pose of the object detected in an RGB-D image in 9-DoF.

In Mask2CAD [14], introduced by Weicheng K. et al., a deep learning network was proposed for detecting an object from an RGB input image and estimating the 3D pose of the object in 5-DoF. The Mask2CAD network segments the region of the object and compares the deep learning features of the segmented object with the features of the CAD models in the embedding space to retrieve the most similar CAD model and estimate the pose of the object. Moreover, Pix3D [15] and Scan2CAD datasets were used in their experiment. In Pix3D, 10,069 images and the annotations of the 3D pose information of objects in the images were used to measure accuracy. Moreover, the performance of the network was tested using 5436 RGB images of Scan2CAD. The results of the experiment for Pix3D showed an intersection over union (IoU) of approximately 0.613, and the average precision ($AP^{mesh}$) was 8.4 for AP50-AP95 [16] of Scan2CAD.

In Pix3D, presented by Sun X et al. [15], a deep learning network was proposed for detecting an object in a single RGB image and estimating the 3D pose of the object in 9-DoF. In addition, a dataset was constructed using RGB images and CAD models for training the network and the annotation information of the 3D pose of objects in the images, and the dataset was published. In particular, in their study, all the objects in the images consisted of only the products sold by IKEA, and the IKEA benchmark dataset was used for the 3D CAD models of the products [17]. The images of IKEA products were crawled from the Internet, and the authors of the paper constructed the dataset using only the images that almost perfectly matched the IKEA products. Hence, the results of the object pose estimation experiment showed an accuracy of approximately 70%.

### 2.3. Industrial Applications of Object Pose Estimation

The 3D pose estimation of objects has several industrial applications, and a bin picking technique is one of the examples. Bin picking is an essential technology for robots to pick up objects or parts and move them to another location or assemble them. Recent technology trends in pose estimation have been identified in the paper by He Z. et al. [1]. Studies on bin picking technology have been ongoing for a long time, even prior to recent deep-learning technology. Research on bin picking technology began with a study that used the feature information of objects in images. Pose estimation using image feature information extracts the feature information of objects detected in the input image primarily using feature point extraction techniques, such as SIFT [18] or SURF [19], and uses the position information of the feature points to estimate the 3D pose of the objects. In addition, a technique that matches the feature points between a 3D CAD model and a 2D object image and uses the PnP algorithm to estimate the 3D pose was introduced [20].

Yu X. et al. [21] recognized complexly arranged objects of various shapes and colors and segmented the regions of the objects. Then, the rotation transformation was calculated as quaternions to transform the CAD coordinate system to the camera coordinate system for each segmented object. In their paper, they proposed a deep learning network called PoseCNN for obtaining the rotation transformation. They also built the fully connected layer of the network in an end-to-end fashion so that quaternions could be regressed. They used the YCB-Video and LINEMOD datasets to evaluate their proposed technique

quantitatively and used both RGB and RGB-D images as the input images. As RGB-D input images were more advantageous for obtaining the rotation information of objects because of the depth and scale information of the objects. Their 6-DoF pose estimation performance was also higher for RGB-D input images.

The DenseFusion 6-DoF pose estimation network proposed by Chen W. et al. [22] used RGB-D images as the input, and RGB and depth features were extracted using a separate CNN network. Then, the features extracted from each image were fused into a global feature using a multilayer perceptron (MLP) to extract features for each image pixel. In the final process, the features of each pixel were used to perform regression on the rotation information of the object with a quaternion model. Their proposed network showed excellent performance and outperformed existing techniques in the ADD-S [18] evaluation.

Yisheng H. et al. [23,24] published two deep-learning-based papers related to pose estimation: PVN3D and FFB6D. As both networks use RGB-D images as the input, they have the drawback of having to obtain the depth information in advance using a color camera and, additionally, a depth camera. A common technique used in both studies is to utilize a CNN and an MLP to find keypoints, which are the most important feature points in RGB images and depth images. ResNet34 [9], trained with general ImageNet [25], was used to extract features from RGB images, and PointNet++ [26] was used to extract features from depth images. In addition, keypoints were extracted using an MLP that shares the result of each CNN as the weight. Here, the keypoints have color and depth information. Hence, a 3D transformation matrix that minimizes the error between these keypoints and the keypoints predefined in the CAD model is calculated. In their study, the FFB6D network does not use an MLP and has a characteristic where all the CNN layers that obtain features from the RGB and depth images share the weights. Then, the rotation and translation matrices are obtained by performing depth-wise concatenation and least squares on the shared features extracted by the CNN.

To develop a bin picking technology for the automation of logistics, Christopher M. et al. [27] rendered the appearance of complexly stacked CAD models of various objects with Blender and generated depth images to use them as the training data for the deep learning network. YOLOv3 was used as the network for detecting objects, and the network trained with CAD models was used to analyze the performance of the detection of objects of various sizes in real images.

### 2.4. RGB Image-Based Pose Estimation

Studies on technologies for estimating the 3D pose of objects in RGB or RGB-D images captured with a camera have been ongoing for a long time [1,2]. As 3D spatial information is required to estimate the pose of an object, performance is generally better when 3D RGB-D images are used than when only 2D RGB images are used. Therefore, further studies have been conducted using RGB-D images as the input—either a learning-based or non-learning-based method was used. However, a separate sensor, such as a depth camera, is required to obtain RGB-D images. Hence, applying this technique to all image-processing fields is difficult. Therefore, multiple learning-based studies have recently been conducted using only RGB images.

Georgios G. et al. [28] proposed a deep learning network that extracts and matches features from an RGB image and a CAD model. If feature point matching between the 2D RGB image and the 3D CAD model is accurate, the 3D pose of the object can be obtained through the RANSAC and PnP algorithms. In addition, VGG was used as the backbone of the network, and a 2048-dimensional descriptor vector was generated for each grid of the image using the keypoint proposal network. Moreover, Georgios G. et al. introduced a more improved deep learning network in [29].

Florian L. et al. [30] also introduced a technique for extracting and matching the feature points of an RGB image and a CAD model using a method similar to [28]. The most difficult problem in matching features between an RGB image and a CAD model is the domain gap. RGB images have natural colors like photographs because they display information

captured with a real camera. However, CAD models contain information generated in a virtual graphic environment, and they do not have sufficient texture information. Hence, the domain gap problem occurs. To solve this problem, in their study, the region of the object was segmented from the RGB image using Mask-RCNN, and the VGG [31] network was used to extract features from the RGB image and the rendered image of the CAD model. To mitigate the domain gap problem in the feature extraction process, the feature vectors were mapped to the joint embedding space to minimize the difference between the vectors of the matching features. In their study, the Pix3D dataset was used for training. Moreover, the proposed pose estimation improves the $AP^{mesh}$ [32] to 37.8 on seen objects and to 17.1 on unseen objects.

Maciej J. et al. [33] compared the similarity between 2D images and 3D CAD models using a Multi-view CNN [34]. Their study does not focus on a technique for estimating the pose of an object in an RGB image. Instead, it focuses on retrieving a CAD model that best matches the object in the RGB image among multiple CAD models. On the other hand, Anan L. et al. [35] introduced a technique for retrieving a CAD model that best matches the object in the input RGB image and estimating the pose using the same multi-view CNN network. To solve the domain gap problem, they trained the network to improve the retrieval performance by projecting the feature vector of the RGB image and the rendered image of the CAD model onto a common vector space.

## 3. Proposed Pose Estimation Pipeline
### 3.1. Object Detection in RGB Images

The 3D pose estimation of an object proposed in this paper includes the rotation and translation transformation relationships between the coordinate system of the CAD models of ShapeNetCore and the coordinate system of the camera that captured RGB images. It also includes the projection transformation from the camera coordinate system to the image coordinate system. The rotation and translation transformations between the 3D coordinate systems have 6-DoF. In addition, in order for the CAD model that has been transformed into the camera coordinate system to match the size of the object in the RGB image when the CAD model is projected onto the camera image space, the size of the CAD model needs to be determined in each of the x, y, and z directions. Hence, additional 3-DoF pose information is required. Consequently, 9-DoF is required. However, this study aims to calculate the pose information between the object in an RGB image and the CAD model using only 4-DoF by minimizing the complexity of pose estimation. Here, 3-DoF is required for rotation transformation, and the translation of the camera and the focal length related to the size of the projected object are calculated in 1-DoF by considering them as one parameter.

First, the rotation and translation transformations, which are external transformation parameters, are relationships between two 3D coordinate systems. Hence, the candidate CAD models need to be determined after determining the category of the object. Therefore, in the first step, the proposed framework detects objects in RGB images. Several studies have been conducted on non-learning- and learning-based object detection methods. Recently, learning-based methods have shown relatively superior performances, and hence, they are commonly used. In this study, DetectoRS [36] was used to detect objects and segment the regions of objects. DetectoRS is a detection and instance segmentation network that uses a feature pyramid network (FPN) [8] as its backbone. It utilizes the FPN repeatedly to extract the features of images more precisely. In this study, the weights pretrained with the Microsoft COCO 2017 dataset were used, and val2017 was used for testing. For the baseline model of the network, ResNet-50 was used as the feature extractor, and a hybrid task cascade was used as the detector. For the learning environment, the epoch was set to 12, and the NVIDIA Titan RTX graphics card was used as the graphics processing unit (GPU). The performance of the trained network was 49.1 for the box AP and 42.6 for the mask AP.

The CAD candidate models were determined from the object category of ShapeNet-Core that matches the class of the object detected by DetectoRS in the RGB input image. Ten candidate CAD models were selected for one category. Section 3.3 describes the candidate CAD models in detail. In addition, a 2D BB was determined with the segmentation information of the region of the detected object, and the image of the object in the BB region of the input RGB image was used for pose estimation.

*3.2. 4-DoF Pose Estimation*

The pose of an object refers to the position and rotation information of the object in 3D space. In Euclidean space, which we generally use, the pose of an object can be expressed by 3D rotation and translation. The rotation can be expressed as a $3 \times 3$ matrix, and the translation can be expressed as a $3 \times 1$ vector. Here, both the rotation and translation matrices have 3-DoF; hence, the 3D pose information has 6-DoF. In addition, as 3D rotation and translation are transformation relationships between two coordinate systems, the coordinate systems must be defined. In this study, the coordinate system of the CAD models of ShapeNetCore is defined as the world coordinate system, and the coordinate system of the camera that captures RGB images is defined as the camera coordinate system, as shown in Figure 3. Rotation from the world coordinate system to the camera coordinate system can be expressed using three rotation angles, as shown in Figure 3. These rotation angles are denoted by azimuth (Azim), elevation (Elev), and inplane rotation (Inpr).

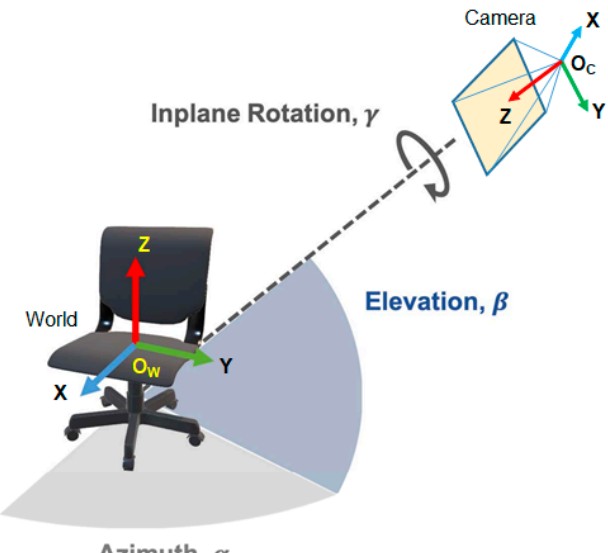

**Figure 3.** Rotation pose representation between the ShapeNet and the RGB camera coordinates.

Let $P_W$ be a point in the 3D CAD model space, and let $P_C$ be this point in reference to the camera coordinate system. Moreover, let $p_B$ be the projection point in the 2D camera image coordinate system of $P_C$. Here, $p_B$ is a 2D coordinate when the BB region of the object in the original RGB input image is considered a new 2D image space. Therefore, 2D translation $t_B$ exists between the image space of the BB region and the coordinate system of the original image space, as shown in Figure 4. Therefore, points $P_W$ and $p_B$ can be expressed using the following perspective projection relation:

$$P_C = [R|T]P_W, \tag{1}$$

$$sp_B = KP_C = \begin{bmatrix} f & 0 & O_x \\ 0 & f & O_y \\ 0 & 0 & 1 \end{bmatrix} P_C. \tag{2}$$

Here, it is assumed that the x and y directions of the focal length, an inner parameter of the camera, are the same. In addition, s is the distance value of the 3D point $\boldsymbol{P_C}$ on the z-axis or the scale value of the projection point $\boldsymbol{P_C}$, which is an arbitrary value that cannot be known.

Suppose that the rotation matrix R from the world coordinate system to the camera coordinate system is calculated first. In other words, if the azimuth, elevation, and inplane rotation are known, the $3 \times 3$ matrix R can be calculated. Subsequently, suppose the CAD model rotated into the camera coordinate system is translated only along the z-axis such that the origin of the world coordinate system of the CAD model intersects the z-axis of the camera coordinate system, as shown in Figure 3. In this case, the $3 \times 1$ translation vector can be expressed as

$$\boldsymbol{T} = \begin{bmatrix} 0 & 0 & t_Z \end{bmatrix}'. \tag{3}$$

Here, []′ is the transpose of the vector. If the above translation vector $\boldsymbol{T}$ is substituted in Equation (1), we obtain

$$\boldsymbol{P_C} = R\boldsymbol{P_W} + \boldsymbol{T}, \tag{4}$$

$$\boldsymbol{P_C} = \boldsymbol{P'_W} + \begin{bmatrix} 0 & 0 & t_Z \end{bmatrix}', \tag{5}$$

$$\mathrm{s}\boldsymbol{p_B} = \boldsymbol{K}\boldsymbol{P_C} = \begin{bmatrix} f & 0 & O_x \\ 0 & f & O_y \\ 0 & 0 & 1 \end{bmatrix} \left( \boldsymbol{P'_W} + \begin{bmatrix} 0 & 0 & t_Z \end{bmatrix}' \right), \tag{6}$$

$$= \begin{bmatrix} f P'_{W_x} + O_x \left( P'_{W_z} + t_Z \right) \\ f P'_{W_y} + O_x \left( P'_{W_z} + t_Z \right) \\ P'_{W_z} + t_Z \end{bmatrix}. \tag{7}$$

We know from Equation (7) that the scale factor s is $P'_{W_z} + t_Z$. Hence, if it is transformed into the 2D projection point, $\boldsymbol{p_B}$ is expressed as follows:

$$\boldsymbol{p_B} = \begin{bmatrix} p_{B_x} \\ p_{B_x} \\ 1 \end{bmatrix} = \begin{bmatrix} f \dfrac{P'_{W_x}}{P'_{W_z} + t_Z} + O_x \\ f \dfrac{P'_{W_y}}{P'_{W_z} + t_Z} + O_y \\ 1 \end{bmatrix} \tag{8}$$

In the above equation, the principal point is the 2D vector $\boldsymbol{t_M} = \begin{bmatrix} O_x & O_x & 1 \end{bmatrix}'$, as shown in Figure 4. Hence, it is not related to the size of the object projected onto the image. Therefore, to project the rotated coordinates $\boldsymbol{P'_W}$ of the 3D CAD model onto the BB image plane so that they exactly overlap with the real object of the RGB image, the parameters related to the change in the size of the 2D image point $\boldsymbol{p_B}$ must be calculated accurately. In Equation (8), the parameters related to the size of the image point $\boldsymbol{p_B}$ and whose values have not been determined are the focal length f and the distance value $t_Z$ from the camera coordinate system to the world coordinate system along the z-axis. However, both unknown parameters need not be calculated. As both the parameters are related to the size of the 2D image space of the object, one parameter can be fixed while the other parameter is varied to obtain a value that achieves the optimal overlap with the object of the BB. Another factor related to the size of the projected object is the coordinates of the CAD model. As the CAD model of ShapeNetCore is not in a real metric space but in a virtual graphic space, its size is an arbitrary value. Therefore, even $P'_{W_z}$, the z value of the rotated CAD model in Equation (8), can be considered an arbitrary value.

Thus, the focal length f of the camera, the z-axis distance $t_Z$, from the origin of the camera coordinate system to that of the world coordinate system, and $P'_{W_z}$—the z value

of the rotated CAD model—in Equation (8) are all unknown scale values. Here, $t_Z$ can be regarded as a constant. Hence, if $t_Z = 2.0$ m, Equation (8) can be simplified as follows:

$$p_B = \begin{bmatrix} p_{B_x} \\ p_{B_x} \\ 1 \end{bmatrix} = \begin{bmatrix} f\frac{P'_{W_x}}{P'_{W_z}+2.0} + O_x \\ f\frac{P'_{W_y}}{P'_{W_z}+2.0} + O_y \\ 1 \end{bmatrix}. \tag{9}$$

According to Equation (9), if $t_Z$ is regarded as an arbitrary constant, the optimal f can be determined by changing the focal length f of the BB image camera and determining how much the image of the CAD model projected onto the BB space overlaps with the region of the actual object. In this study, the object in an RGB image is detected, and the region of the object is segmented from the image. Then, the RGB image of the BB region is used as the input to estimate the pose. Hence, the 3D pose information of the object can be obtained by calculating both the 3-DoF rotation and the focal length f that makes the projected region of the CAD model overlap with the object region the most. Thus, the complete pose information is determined in 4-DoF. Previous studies had to solve highly complex problems of 6-DoF, including the rotation and translation, or 9-DoF if including the scale of the CAD model. In comparison, the proposed technique has the advantage of minimizing the complexity of the pose estimation problem to 4-DoF.

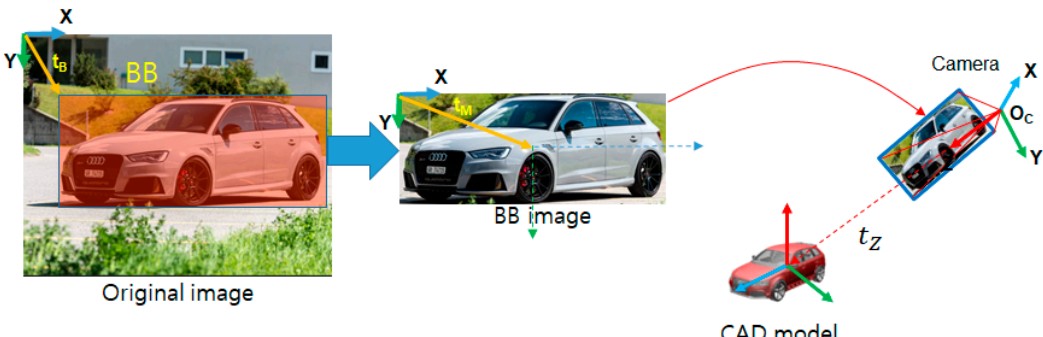

**Figure 4.** Definition of the translation 't$_z$' from the camera to the CAD model coordinates.

### 3.3. Estimation of Rotation and Focal Length

The previous section provided the theoretical background of the 4-DOF-based object pose estimation proposed in this paper. This section describes the method of calculating 4-DoF, that is, 3-DoF for rotation and 1-DoF for the focal length f. Estimating the rotation between the pose of the object detected in the RGB image based on the camera coordinate system and the ShapeNetCore world coordinate system of the matching CAD model has a high complexity of 3-DoF. Previous related studies also used the feature values for a visual change owing to the rotation of the object. For learning-based methods, previous studies have adopted the method of learning these features using a deep learning network. However, as observed in FFB6D, which is a recent learning-based pose estimation method that uses RGB images as the input, even learning all the visual features of the 3-DoF rotation values for an object class requires a large amount of training data and deep CNN layers. Hence, FFB6D is trained on one class and performs pose estimation for that class. In the case of the existing methods using RGB-D images as the input, additional depth information is available. Hence, these methods simultaneously learn the poses of objects from multiple classes and use them. However, in this study, only RGB images are used as the input, and hence, it is difficult to utilize them.

In this study, the 3-DoF pose is estimated using only the RGB input images of objects of several classes. Hence, the existing method based on the image features of the objects is not used. Instead, PoseContrast, which is a learning-based method that can estimate the pose

regardless of the class of the object, is used. PoseContrast is a learning method that uses pose-aware contrast loss to consider only the pose of the object and not its class. The pose contrast loss of the object in the RGB image is based on SimCLR, which applies different types of data augmentation to the image regardless of the class of the object. Then, the feature representations of positive pairs are trained so that the distance between the feature vectors decreases, and the feature representations of negative pairs are trained so that the distance between the feature vectors increases. Based on this principle, PoseContrast learns pairs with different poses as negative pairs and pairs with similar poses as positive pairs.

Three datasets—Pascal3D+, ObjectNet3D, and Pix3D—were used to train PoseContrast. The entire dataset comprises 80,696 training images and 42,306 validation images for 121 classes. Each dataset contains Euler angle information (azimuth, elevation, inplane rotation) as the ground truth for learning the poses. The Adam optimizer was used in an end-to-end manner for training, and the batch size was set to 32. The initial learning rate was $10^{-4}$ at 15 epoch, and the training was conducted for approximately two hours using the V-100 16G GPU. In addition, MoCo v2 (pretrained ResNet-50) was used as the image encoder network.

After the rotational pose of the object is estimated, the method for estimating the camera focal length f of 1-DoF is as follows. To project the CAD model onto the 2D BB image space, the model needs to be expressed as points, and Equations (4)–(8) need to be applied. However, all the CAD models of ShapeNetCore are saved in a mesh format. Therefore, it is difficult to project the vertices of the mesh model onto the image plane and compare the degree of overlap between the projected model and the image information of the actual object. Therefore, in this study, the CAD model expressed in a mesh format was transformed into point clouds, which were then projected onto the BB image. Then, the degree of overlap between the distribution of the point clouds and the segmented region of the object was determined using the IoU. As shown in Figure 5, the CloudCompare [37] software was used to transform the mesh model into point clouds. The mesh of the CAD model consists of vertices. However, as shown in Figure 5b, the number of points is insufficient to calculate the IoU. Therefore, the mesh surface was sampled as points, as shown in Figure 5c. In addition, numerous points were projected onto the BB image plane, and an f value with an IoU close to 1.0 was selected. After projecting the point cloud data of the model onto the BB image space using the projection transformation matrix in Equation (2), a rectangular space was defined with the maximum and minimum values of x and y of the projected points. Then, the IoU between this rectangular space and the BB rectangular space was used to determine the most suitable focal length.

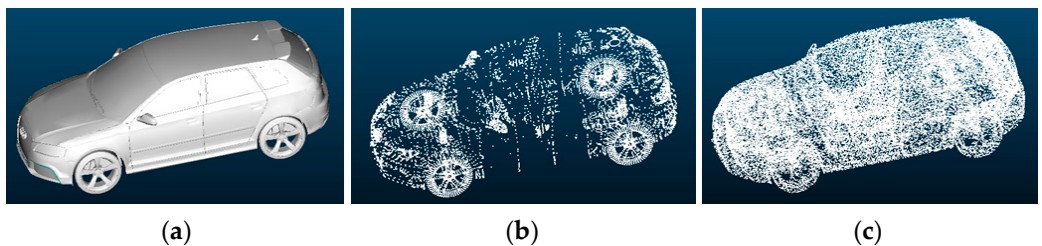

| (a) | (b) | (c) |

**Figure 5.** An example of the graphical representation of a CAD model. (**a**) Mesh, (**b**) mesh vertices: 17,815, and (**c**) resampled points: 60,000.

### *3.4. Retrieval of a CAD Model and Similarity Measurement*

The CAD model whose shape best matches the object detected in the RGB input image was retrieved from ShapeNetCore. ShapeNetCore consists of 51,300 CAD models classified into 55 categories. Each category is further divided into several subcategories. For example, the category "chair" has 23 subcategories and consists of 6778 models. Therefore, the class of the object detected in the RGB image can be regarded as one type of category in ShapeNetCore. If the class of the object detected in the image is "chair," the model with the best matching shape among 6778 chairs in the "chair" category needs to be retrieved

from ShapeNetCore. However, such a retrieval task is difficult, mainly because of the domain gap problem. The CAD models of ShapeNetCore are 3D mesh data generated with a graphic technology. The texture of the CAD models is also generated with the graphic technology. Hence, there is a significant difference between the CAD model and the actual appearance. Even if a 2D rendered image of a CAD model is generated for a comparison with the image features of the object detected in the input image, the 2D rendered image also has noticeable graphic texture characteristics. Hence, there is a significant difference between the 2D rendered image and the actual image.

In retrieving CAD models, the number of candidate CAD models is another problem, along with the domain gap problem. As mentioned in the earlier example, the "chair" category, which is the most common object among indoor objects, has 6778 CAD models. In the case of the "table" category, there are 26 subcategories and 8436 models. Hence, retrieving a CAD model identical to the object in the actual image is a difficult task. Hence, previous studies using the ShapeNet dataset did not use all the CAD models but used only a small number of models. As another example, only 90 CAD models were used out of 759 CAD models in the IKEA dataset [17] in Pix3D [15]. As Pix3D uses nine categories, ten CAD models were used for each category on average.

In this study, we also selected ten categories in ShapeNetCore. For each category, ten CAD models with commonly encountered shapes were selected. In other words, ten categories and one hundred CAD models were used in the experiment, as shown in Figure 6. As shown in the figure, CAD models with different shapes were selected within each category for easier classification. However, as evident in the example of the all categories, the shapes of the candidate CAD models were still similar even though we manually selected the candidate models. Hence, retrieving a model that best matches the object in the input image among the candidate models is a difficult task.

As explained earlier, the category of the object detected in the RGB input image was determined, and the CAD model with the same category was retrieved from ShapeNetCore. In general, to learn a category called "car" with the current learning-based deep learning technology, a dataset is constructed with a large volume of car image data with various shapes, and this dataset is used to train the network. For example, even in training datasets, such as ImageNet [25], Microsoft COCO dataset [16], and YOLOv3, categories related to "car" are only classified into categories with a broad meaning, such as "vehicle" and "truck." However, the CAD models required in this study must have almost the same shape as the object in real images. The technique developed by us is utilized in the XR field. Hence, when the 3D CAD model is projected onto the 2D image space with the pose information of the actual object, the projected model must align with the region of the object accurately. This indicates that the shape of the detected object must match the shape of the model of ShapeNetCore almost exactly. Therefore, a model that best matches the object detected in the image must be retrieved from the ten candidate models selected by us for each category.

This operation can be considered one of the 2D–3D model retrieval methods. As we determined the approximate 3D rotation information of the object detected in the image using PoseContrast, we intend to solve the 2D–3D retrieval problem as a 2D–2D matching problem. The rotation information (Azim, Elev, Inpr) of the object obtained using PoseContrast in Section 3.3 and Blender Python API [38] are used to save the shape of a candidate CAD model rotated into the camera coordinate system as a 2D image. If the PoseContrast result can determine the approximate rotation information of the object, the candidate model can be rotated to the viewpoint of the camera, and a 2D image of an appropriate size can be generated. The image generation result can be regarded as the image of a candidate model with the same rotation information as the actual object. Moreover, we determine the final model by measuring the similarity between the BB image that only includes the region of the object in the input image and the 2D image of the candidate model generated using Blender Python API.

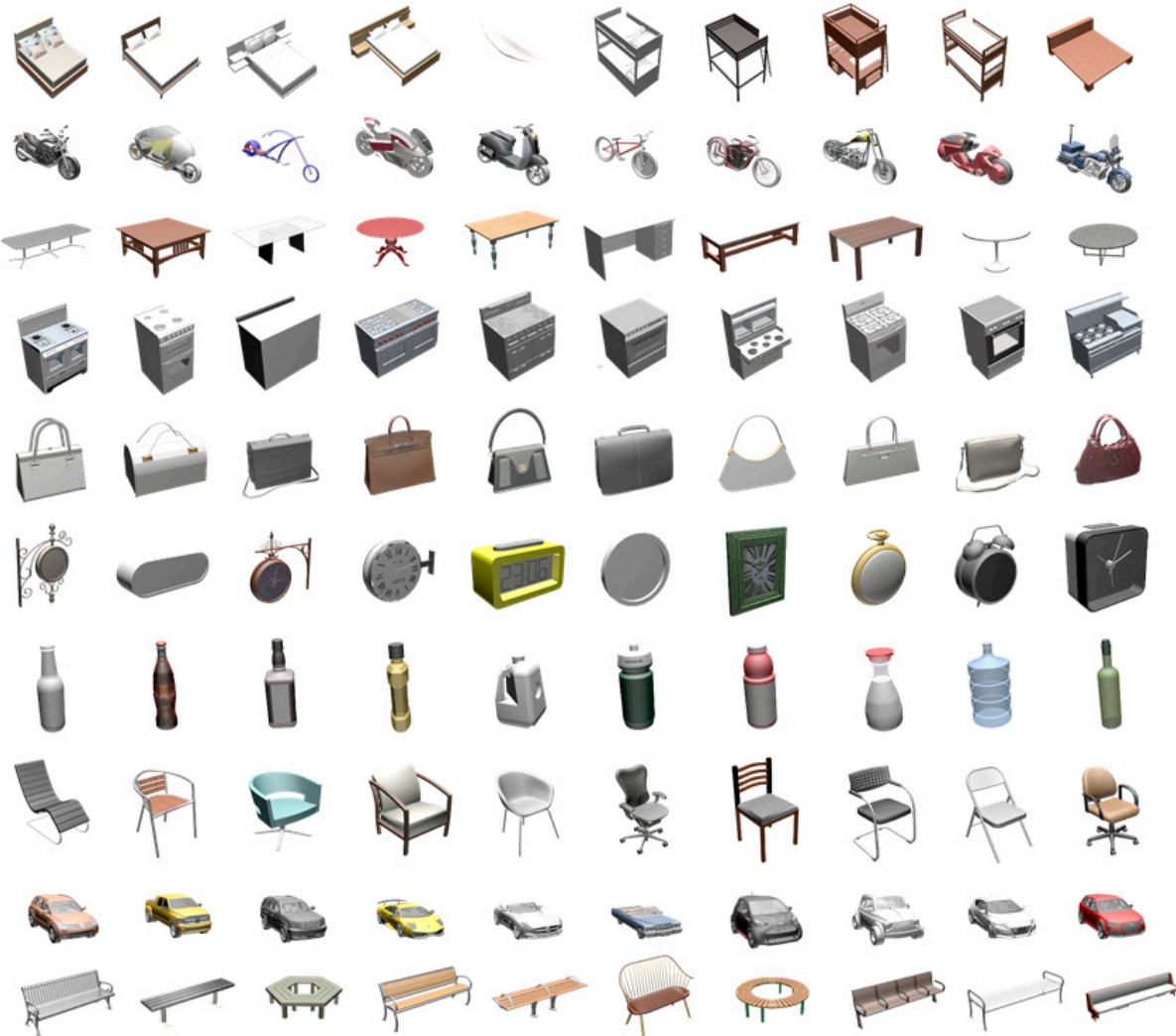

**Figure 6.** Ten ShapeNetCore categories and candidate CAD models used in the experiment. From the top, *Bed, Motobike, Table, Oven, Bag, Clock, Bottle, Chair, Car,* and *Bench* (The italic type means the test object's name).

Measuring the similarity between the actual image and the rendered image of the CAD model is a difficult problem owing to the domain gap. The domain gap is the most difficult problem in current learning-based object detection or retrieval methods [39]. This study does not deal with the domain gap problem directly. Instead, the final CAD model is retrieved using the 2D–2D similarity measurement. In addition, the local features, rather than global features, are compared between the object and the model to minimize the domain gap problem in the similarity measurement process. As the category of the object has already been determined in the object detection phase, a method that can distinguish different local features among 10 candidate models with the same category is more appropriate than to compare the global features among the candidate models again.

The local feature information is used to measure the similarity between the RGB image of the actual object (within the BB region) and the 2D rendered image of the candidate CAD model. In this study, the DELF [40] features are used to extract the local features of images. DELF uses a learning-based CNN with the descriptor of image features required for large-scale image retrieval. In a previous study [40], a two-phase process was performed for large-scale image retrieval. We modified the image retrieval pipeline, which is the second phase in DELF, and used it to measure the 2D–2D similarity. In the image retrieval pipeline of DELF, the nearest neighbor is determined among the feature descriptors between the

query image and all the images in the database. Subsequently, the matching points with each database image are collected separately. Then, geometric verification is performed using RANSAC [41] to determine the best-matching database image.

In this study, DELF was used to obtain and match all the features between the RGB query image and the rendered images of ten candidate models. As the 3D pose of the object was already estimated approximately from the query and rendered images—which are used to measure the similarity—the pose of the object viewed in the image is almost similar. Therefore, it is assumed that there is only small difference in rotation or scale between the query and rendered images. Consequently, the 2D affine transformation model was used in this study for the geometric verification of image feature matching. The final matched candidate model was determined using the affine transformation model in the order of the most inliers.

Suppose the $3 \times 3$ affine matrix $A_{ij}^k$ between the RGB query image and $k$-th rendered CAD image is determined by matching the DELF features between them. And suppose $I_{ij}$ is the same size identity matrix. Here, $i$ and $j$ are indices of the matrix elements. In addition, let $N_{inl}^k$ be the number of feature inliers between them. We use following three criteria for image similarity measurement $\rho_k$ for the $k$-th rendered CAD image. The final candidate is selected which has the maximum $\rho_k$. We demonstrated through the experiment that the affine transformation model is sufficient to perform the geometric verification between the query and rendered images.

- Similarity measure 1 (SM1):

$$\varepsilon_k = \sum_{i,j} \left| \frac{A_{ij}^k}{\max\left(A_{ij}^k\right)} - I_{ij} \right| \tag{10}$$

$$\rho_k = N_{inl}^k \; of \; k\text{-}th \; rendered \; image \; among \; five \; smallest \; \varepsilon_k$$

- Similarity measure 2 (SM2):

$$\rho_k = N_{inl}^k \tag{11}$$

- Similarity measure 3 (SM3):

$$\varepsilon_k = \sum_{i,j} \left| \frac{A_{ij}^k}{\max\left(A_{ij}^k\right)} - I_{ij} \right| \tag{12}$$

$$\rho_k = (1 - \alpha(\text{sigmoid}(\varepsilon_k)) \times N_{inl}^k \; (\alpha = 0.35)$$

## 4. Experiments

### 4.1. Object Detection and Segmentation

To evaluate the performance of the proposed method, we use total ten indoor and outdoor objects, *Bed, Motobike, Table, Oven, Bag, Clock, Bottle, Chair, Car, and Bench* as shown in Figure 7. One common property of the test objects is that they are nonsymmetric. Symmetric objects such as bottle or cup is not used to reduce the pose ambiguity problem. In each object category, we collect 100 RGB images from the Internet. When collecting the images, we try to collect such images whose color and shape are similar with those of the corresponding CAD models as much as possible.

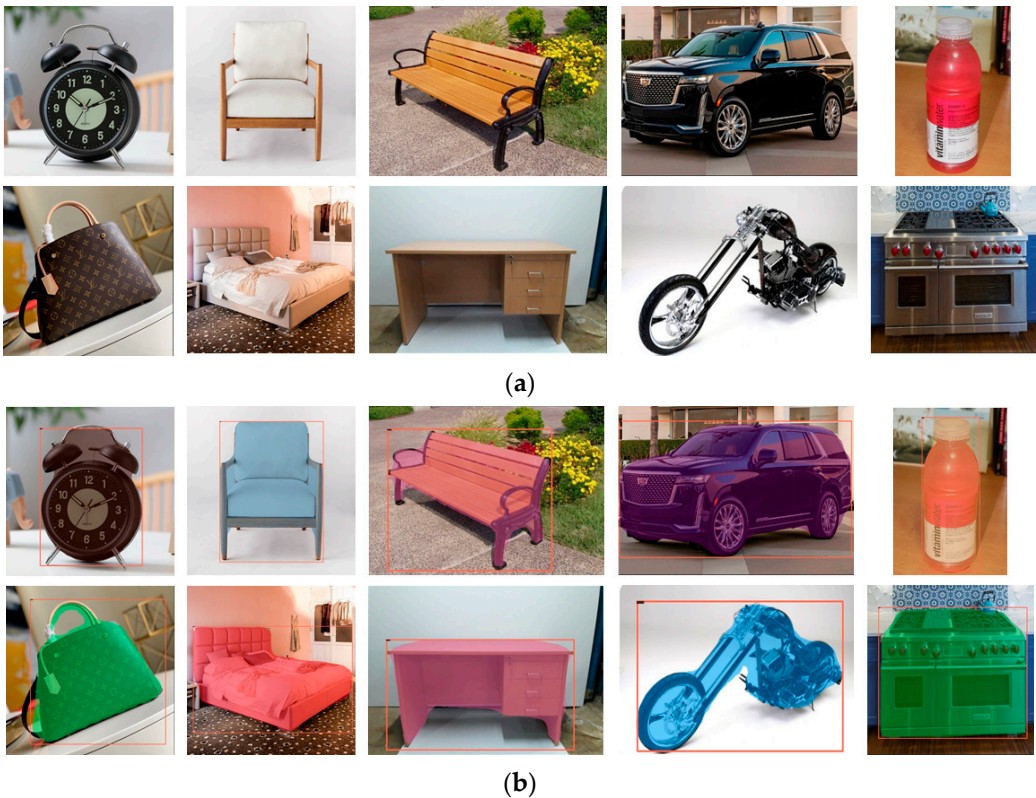

**Figure 7.** Example images of ten test objects. Clockwise direction from the top-left: Clock, Chair, Bench, Car, Bottle, Bag, Bed, Desk, Motobike, and Oven. (**a**) RGB input images (**b**) Detection and semantic segmentation results.

In each object category, we have total ten CAD candidates. In fact, in each object category, there is a large number of CAD models in ShapeNetXCore. For example, in the *Clock* category, there are three sub-categories and 651 CAD models in total. Thus, there are large overlaps in terms of similarity of color and shape. One of the purposes of our study is finding the matching pair between the real and the CAD model. Therefore, it needs to be the CAD models distinguishable in terms of color and shape. In each object category, we select ten CAD models which are commonly available in daily life and have different color and shape.

*4.2. Pose Estimation and Similarity Measurement*

In the previous section, the details of the object category and segmentation are described. As the next step, the rotational pose of the object is estimated. In Figure 8, the bounding box images of four sample objects are shown, *Car, Bench, Bag*, and *Clock*. At the right side of the image, ten CAD models of the detected categories are shown. In this figure, the rotational pose of the CAD models is determined with the results from PoseContrast. The images of the CAD models are rendered by Blender Python API. As shown in this figure, the pose of the CAD models is very close to that of input real object. Appearance of the CAD models look different so that image similarity measurement can recognize the instance of the models.

Figure 9 shows DELF feature matching results of sample four object categories between the RGB input and CAD candidates. In each result, the left is RGB query and right is rendered CAD image. The DELF features are detected in both RGB input images and the rendered images. The blue-colored lines show the inlier matching features between the two images. Because the pose of the real objects and CAD models are very similar, the feature matching results implies that there is 2D affinity relationship between the real and CAD images. Based on the number of inliers, only top five matching results are shown and

the left one is the best match. Among them, the red-colored box in each category means ground truth CAD model.

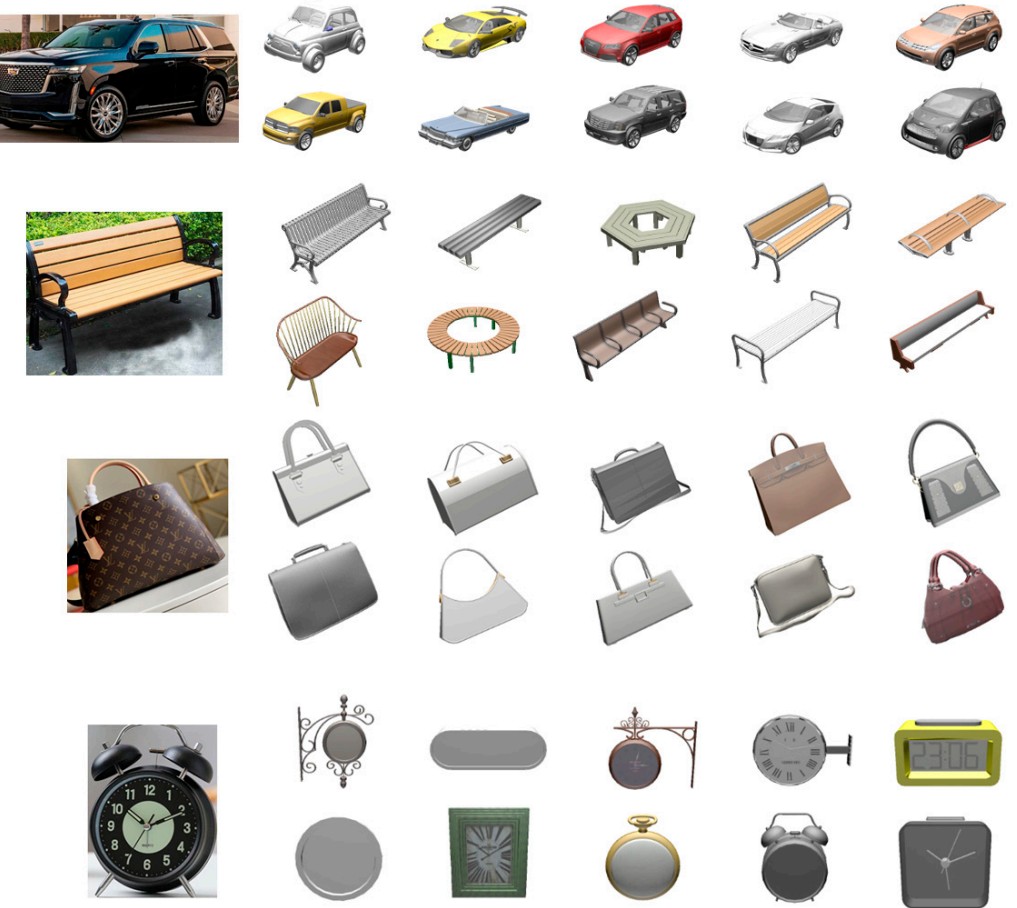

**Figure 8.** Rendering images of CAD candidates using the pose estimation results. From the top to the bottom, results of four test objects are shown, *Car, Bench, Bag*, and *Clock*. In each RGB image at the left, the rendered images of the ten CAD candidates are shown in the right. Comparing the pose of the objects in both RGB and rendered images, it is found that the rotational pose of the objects is successfully estimated (Due to the space limit, results of only four objects are shown).

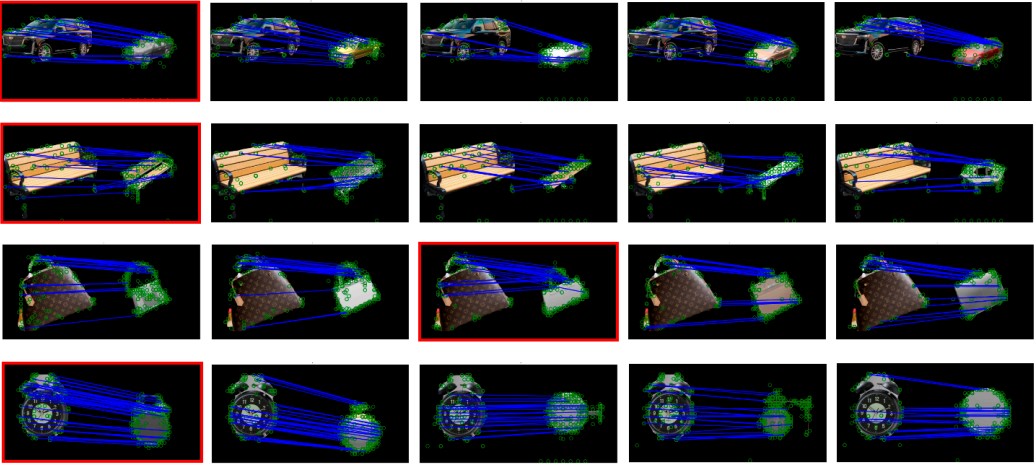

**Figure 9.** From the top, DELF feature matching results of CAR, Bench, Bag, and Clock. In each row, the leftmost CAD model is determined as the best similar object with the RGB object.

### 4.3. Performance Analysis of Similarity Measurement

Tables 1–3 shows similarity matching results with three different measure criteria. As mentioned before, we use ten object categories and each category has 100 RGB test images, total 1000 RGB query images are used for experiments. However, at the object detection and pose estimation steps, some query images are rejected because of detection and estimation failure. Thus, in the similarity measurement experiment, total 596 query images are used.

**Table 1.** CAD model retrieval results using the similarity measurement SM1.

| Category | 1-st Rank | 2-nd Rank | 3-rd Rank | Top-3 |
|---|---|---|---|---|
| *Bag* (19) | 7 | 2 | 0 | 9 |
| *Bed* (24) | 4 | 7 | 3 | 14 |
| *Bench* (63) | 38 | 10 | 1 | 49 |
| *Bottle* (77) | 47 | 14 | 8 | 69 |
| *Car* (80) | 33 | 11 | 8 | 52 |
| *Chair* (80) | 45 | 21 | 3 | 69 |
| *Clock* (59) | 27 | 12 | 4 | 43 |
| *Motorbike* (76) | 37 | 8 | 8 | 53 |
| *Oven* (79) | 34 | 10 | 10 | 54 |
| *Table* (39) | 18 | 2 | 2 | 22 |
| Total (596) | 290 | 97 | 47 | 434 |
| Success rate (%) | 48.65 | 16.27 | 7.88 | 72.81 |

**Table 2.** CAD model retrieval results using the similarity measurement SM2.

| Category | 1-st Rank | 2-nd Rank | 3-Rd Rank | Top-3 |
|---|---|---|---|---|
| *Bag* (19) | 7 | 3 | 3 | 13 |
| *Bed* (24) | 6 | 7 | 3 | 16 |
| *Bench* (63) | 39 | 16 | 2 | 57 |
| *Bottle* (77) | 41 | 15 | 14 | 70 |
| *Car* (80) | 33 | 16 | 6 | 55 |
| *Chair* (80) | 45 | 20 | 2 | 67 |
| *Clock* (59) | 26 | 13 | 4 | 43 |
| *Motorbike* (76) | 39 | 8 | 6 | 53 |
| *Oven* (79) | 33 | 9 | 12 | 54 |
| *Table* (39) | 20 | 4 | 1 | 25 |
| Total (596) | 289 | 111 | 53 | 453 |
| Success rate (%) | 48.48 | 18.62 | 8.89 | 76.00 |

**Table 3.** CAD model retrieval results using the similarity measurement SM3.

| Category | 1-St Rank | 2-Nd Rank | 3-rd Rank | Top-3 |
|---|---|---|---|---|
| *Bag* (19) | 6 | 1 | 4 | 11 |
| *Bed* (24) | 6 | 5 | 5 | 16 |
| *Bench* (63) | 39 | 15 | 1 | 55 |
| *Bottle* (77) | 41 | 14 | 15 | 70 |
| *Car* (80) | 32 | 14 | 6 | 52 |
| *Chair* (80) | 43 | 20 | 3 | 66 |
| *Clock* (59) | 25 | 14 | 5 | 44 |
| *Motorbike* (76) | 38 | 7 | 7 | 52 |
| *Oven* (79) | 33 | 9 | 12 | 54 |
| *Table* (39) | 16 | 6 | 3 | 25 |
| Total (596) | 279 | 105 | 61 | 445 |
| Success rate (%) | 46.81 | 17.61 | 10.23 | 74.66 |

In an ideal case, the similarity measurement finds the real matching CAD model which yields the highest similarity score. However, due to the texture quality of the CAD models, about 48% of the input finds the matching CAD models. Thus, we analyze the similarity measure up to the top-3 similar CAD images as shown in the tables. As shown in Tables 1–3, SM2 measurement shows the best performance, which simply counts the number of matching inliers. However, in terms of the top-first model retrieval, SM1 is the best measurement.

### 4.4. Comparison with a Triplet Loss Learning Method

The performance of the proposed method is also compared with a learning based CAD model retrieval method [30]. In the deep learning architecture in [30], a triplet loss learning network is employed for CAD model retrieval. This method also retrieves the top-n matching CAD rendering images for candidate models, and a key point matching is used to determine the final best matching model. We also employ the triplet loss learning network for comparison with our method as shown in Table 4.

The triplet loss learning network in [30] composed of two VGG encoders to train the positive and negative features between the RGB and CAD rendering images in a feature embedding space. Here, the features from the RGB images become the anchor in the feature space. For each RGB input image, we use one matching CAD image as the positive label and ten mis-matching CAD images as the negative labels. The mis-matching CAD images are randomly selected from all categories. As the encoder, the VGG16 model is used and it was trained with ImageNet2014. For training, total 592 RGB images are used and the batch size is set to 6. As the optimizer, SGD is used and the learning rate is set to $2 \times 10^{-6}$. To test the triplet loss network, total 100 RGB images are used which have not been included in the learning. Furthermore, to improve the performance of the triplet loss network, the negative mining strategy is also used. For negative mining, the CAD images of the bottom-7 negative images are reused for training the network.

**Table 4.** Comparison of CAD retrieval performance with triplet loss network [30]. (100 untrained RGB images are teste.d).

| Methods | 1-st Rank | 2-nd Rank | 3-rd Rank | Top-3 |
|---|---|---|---|---|
| Triplet loss learning [30] (w/o negative mining) | 31 | 18 | 19 | 68 |
| Triplet loss learning [30] (w/negative mining) | 29 | 25 | 17 | 71 |
| Proposed (SM1) | 56 | 13 | 4 | 73 |
| Proposed (SM2) | 56 | 16 | 6 | **78** |
| Proposed (SM3) | 51 | 15 | 12 | **78** |

Figure 10 shows the final image alignment results using the proposed 4-DoF pose estimation. Onto the real image space, the matched CAD model, which is the top-first CAD model from the similarity measure, is projected and overlapped. To project the matched CAD model, we convert the mesh structure of the CAD model to point clouds using CloudCompare. This alignment results show that the pose of the matched CAD model is reasonably accurate for the application to the extended reality. As shown in the figure, there is still overlap errors in some objects, which is caused by shape mismatch between the real object and the CAD model. For example, the result of *Bag* object show that the matched CAD model couldn't perfectly overlap with the real object area. This is due to the fact that there is no perfectly matching shape among our CAD candidates.

We present Figure 10 only for showing examples of CAD model alignment in the input images, not showing the performance measurement of the proposed pose estimation. For quantitative measurement of the alignment performance, the shape of the CAD model should be same with the real object in the image so that the alignment error is measured in

the image space. However, this paper uses the Internet database for the most-similar CAD retrieval, thus the quantitative alignment error analysis is not addressed.

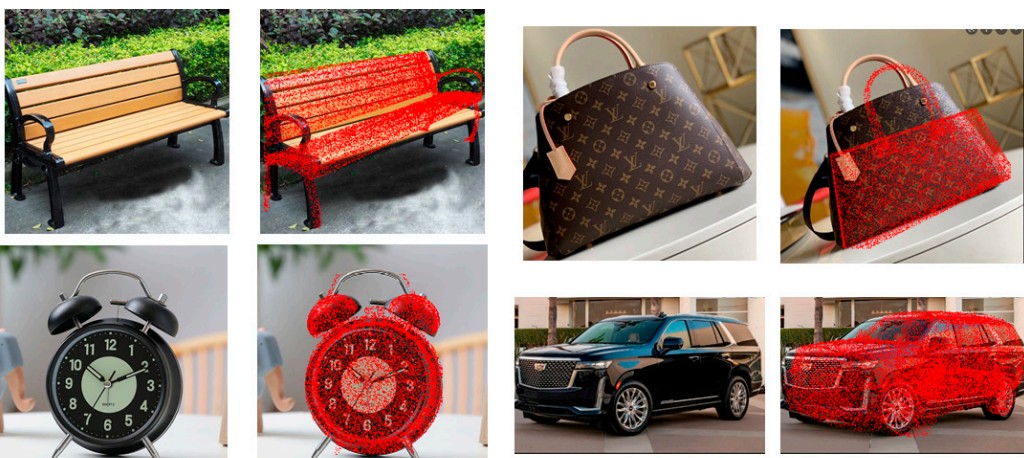

**Figure 10.** The matched CAD model is projected to the 2D image plane with the 4-DoF pose information. 4-DoF pose of the Clock and Car objects are very closely matched with the real objects. The Bench model projection is not perfectly matched with real object area due to a small pose error. The projection error of the Bag object CAD is caused by the shape mismatch between the real object and the CAD model.

## 5. Discussion

The goal of the proposed pipeline is to estimate the 3D pose of a real object in a single image only with the 4-DoF annotation parameters to a matching CAD model. To achieve the goal, the proposed pipeline detects objects of various classes in RGB images, retrieves CAD candidates with the same shape, estimate the rotational pose of the objects, and performs the similarity measurement to find the base matching 3D model. The rotational pose is represented only with 3-DoF parameters and the focal length of the camera for alignment of the model to the object is represented only with 1-DoF parameter. Thus total 4-DoF pose parameters are enough to annotate the pose of the CAD model relatively with the object area in an RGB image.

In this study, ten CAD models were arbitrarily selected for each class of the object, and the similarity between the rendered image of the rotated CAD model and the RGB image was measured to determine the optimal CAD model. DetectoRS was trained to detect the object and its class, and PoseContrast was used for rotation among the detected object pose information. The similarity between the rendered images of the 10 candidate CAD models with the same class as the class of the detected object was measured by matching the DELF feature to minimize the domain gap problem.

From experiments, we find that only 4-DoF pose annotation parameters are enough to project the CAD model to the image space to align the shape of the model with the object region. The proposed pose estimation framework can be used to several graphic-to-image fusion applications such as Augmented or Extended Reality.

**Author Contributions:** Conceptualization, S.-Y.P.; methodology, software, validation, formal analysis, C.-M.S., W.-J.J. and S.P.; investigation, S.-Y.P., C.-M.S., W.-J.J. and S.P.; writing—original draft preparation, S.-Y.P.; writing—review and editing, S.-Y.P.; supervision, S.-Y.P.; project administration, S.-Y.P.; funding acquisition, S.-Y.P. All authors have read and agreed to the published version of the manuscript.

**Funding:** This research was by Institute of Information & communications Technology Planning & Evaluation (IITP) grant funded by the Korea government (MSIT) (No. 2021-0-00320, Manipulation and Augmentation for XR in the Real-World Environment).

**Institutional Review Board Statement:** Not applicable.

**Informed Consent Statement:** Not applicable.

**Data Availability Statement:** The data presented in this study are available on request from the corresponding author.

**Conflicts of Interest:** The authors declare no conflict of interest.

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
