# Peer review of "Relative Pose Estimation between Image Object and ShapeNet CAD Model for Automatic 4-DoF Annotation"

_applsci, doi:10.3390/app13020693_

Round 1
Reviewer 1 Report
Using an RGB image to estimate the three-dimensional (3D) pose of a real object is an interesting and difficult topic. This research proposes a new pipeline, which can estimate and represent the pose of objects in RGB images by DOF annotation of matched CAD models.
This study shows that only 4-DOF annotation parameters between real objects and CAD models are sufficient to promote the projection of CAD models into RGB space for image graphic applications, such as extended reality. In the experiment, the performance of this method is analyzed using real ground data, and compared with the triple loss learning method.
1. For the introduction part of the manuscript, in order to enable readers to better understand the advantages of the proposed method, it is recommended to add some content to the introduction part of the manuscript and summarize the contributions of the manuscript.Please refer the recent papers: Chenggang Yan, Biao Gong, Yuxuan Wei, Yue Gao, “Deep Multi-View Enhancement Hashing for Image Retrieval”, IEEE Transactions on Pattern Analysis and Machine Intelligence, 2020.Chenggang Yan, Zhisheng Li, Yongbing Zhang, Yutao Liu, Xiangyang Ji, Yongdong Zhang, “Depth image denoising using nuclear norm and learning graph model”, ACM Transactions on Multimedia Computing Communications and Applications 2020.Chenggang Yan, Yiming Hao, Liang Li, Jian Yin, Anan Liu, Zhendong Mao, Zhenyu Chen, Xingyu Gao, “Task-Adaptive Attention for Image Captioning”, IEEE Transactions on Circuits and Systems for Video Technology, 2021. Chenggang Yan, Tong Teng, Yutao Liu, Yongbing Zhang, Haoqian Wang, Xiangyang Ji, “Precise No-Reference Image Quality Evaluation Based on Distortion Identification”, ACM Transactions on Multimedia Computing Communications and Applications 2021.Chenggang Yan, Lixuan Meng, Liang Li, Jiehua Zhang, Jian Yin, Jiyong Zhang, Zhan Wang, Bolun Zheng, “Age-Invariant Face Recognition By Multi-Feature Fusion and Decomposition with Self-Attention”, ACM Transactions on Multimedia Computing Communications and Applications 2021
1. The manuscript determines categories and object regions from a single RGB image, and segments them. Secondly, only the segmented object region is used to estimate the 3-DOF rotation attitude of the object through the learned attitude contrast network.
2. In this study, 10 CAD models are randomly selected for each type of object, and the similarity between the rendered image of the rotating CAD model and the RGB image is measured to determine the optimal CAD model.
Author Response
Dear Reviewers,
Thank you for your valuable comments on our manuscript, ”Relative Pose Estimation between Image Object and ShapeNet CAD Model for Automatic 4-DoF Annotation” submitted to ‘Applied Science’ journal. We have revised the manuscript according to all reviewer’s comments. In below, more revision details are shown in each comment from the reviewers.
Reviewer #1 :
Comments and Suggestions for Authors
Using an RGB image to estimate the three-dimensional (3D) pose of a real object is an interesting and difficult topic. This research proposes a new pipeline, which can estimate and represent the pose of objects in RGB images by DOF annotation of matched CAD models.
This study shows that only 4-DOF annotation parameters between real objects and CAD models are sufficient to promote the projection of CAD models into RGB space for image graphic applications, such as extended reality. In the experiment, the performance of this method is analyzed using real ground data, and compared with the triple loss learning method.
- For the introduction part of the manuscript, in order to enable readers to better understand the advantages of the proposed method, it is recommended to add some content to the introduction part of the manuscript and summarize the contributions of the manuscript.
Answer: In the end of Introduction, we add the summary of the proposed method as follows:
- The 4-DoF pose estimation pipeline from a single RGB image is proposed.
- The 4-DoF pose annotation database can be generated to align the CAD model of a real object in the image plane.
- Three image similarity criteria are proposed to match deep features between rendered CAD and real object images.
- Please refer the recent papers: Chenggang Yan, Biao Gong, Yuxuan Wei, Yue Gao, “Deep Multi-View Enhancement Hashing for Image Retrieval”, IEEE Transactions on Pattern Analysis and Machine Intelligence, 2020.
Answer: Suggested reference papers are not related with our study. Thus, the references are not added.

Reviewer 2 Report
This manuscript proposes a method to estimate and represent the pose of an object in an RGB image only with the 4-DoF annotation to a matching CAD model. The topic is interesting, and the method has certain innovation. Some comments are listed as follows:
1. The proposed method has the assumption that there is only small difference in rotation or scale between the query and rendered images. The reviewer wonder what impact will have on results without this assumption.
2. The manuscript only presents qualitative results of the proposed 4-DoF pose estimation method as Fig. 9 illustrates. However, quantitative results are required to be presented.
3. The reviewer thinks that it is better to present comparison of results of pose estimation method with other existing methods.
4. There are some minor typos and grammar errors.
(1) For the text from Figure 2 to Figure 6, the description of figure number is incorrect. For example, in page 7 line 301, Figure 3 should be Figure 2.
(2) In the definition of SM1, Iij should be explained.
(3) The equations of SM1, SM2 and SM3 should be numbered.
(4) There is no conjunction between sentences.
Page 18 Line 655, There is still overlap errors in some objects, it is caused ...
(5) Page 13 Line 559, is to finding
(6) Page 14 Line 576, next, step
Author Response
Dear Reviewers,
Thank you for your valuable comments on our manuscript, ”Relative Pose Estimation between Image Object and ShapeNet CAD Model for Automatic 4-DoF Annotation” submitted to ‘Applied Science’ journal. We have revised the manuscript according to all reviewer’s comments. In below, more revision details are shown in each comment from the reviewers.
Reviewer #2 :
Comments and Suggestions for Authors
This manuscript proposes a method to estimate and represent the pose of an object in an RGB image only with the 4-DoF annotation to a matching CAD model. The topic is interesting, and the method has certain innovation. Some comments are listed as follows:
- The proposed method has the assumption that there is only small difference in rotation or scale between the query and rendered images. The reviewer wonder what impact will have on results without this assumption.
Answer: I think the reviewer misunderstands the proposed pose estimation method. In an input image, there is no assumption that the rotation or scale of the image object should similar to the rendered CAD model image. Instead, in any pose of the object in an image, we crop the object image region and the pose of the object is estimated by PoseContrast. Then, the pose of the CAD candidates is aligned with the estimated pose and candidates are rendered to images. That’s why the similarity measurement uses similar pose images, object and rendered CAD images.
- The manuscript only presents qualitative results of the proposed 4-DoF pose estimation method as Fig. 9 illustrates. However, quantitative results are required to be presented.
Answer: (I think the reviewer tells about the Figure 10, in the original PDF version, the figures are incorrectly numbered.) The quantitative pose estimation analysis is shown in Table 2~4. Figure 10 shows just alignment examples of CAD models with the input image objects. The figure also shows the proposed method can provide reasonable 4-DoF annotation for pose estimation. It is not appropriate doing any quantitative analysis from the results of alignment of Figure 10. The main reason is that the 3D shape of CAD models is not exactly same with the image objects as shown in Figure 10. If the 3D shape is different, the alignment error analysis in the image space is not meaningful.
With this regard, we add additional explanations of the reason of presenting Figure 10 in the end of Section 4.4. The revised parts are written with red-colored texts.
- The reviewer thinks that it is better to present comparison of results of pose estimation method with other existing methods.
Answer: In the original manuscript, comparison with another existing method has been done with reference [31], the triplet loss learning network. The triplet loss learning network has been trained with our own training database, 592 RGB images, which are used in our DELF-based similarity measurement. With and without negative mining is also compared as shown in Table 4. Comparison with the other existing methods could not been done because of only 5 days of revision time limitation.
- There are some minor typos and grammar errors.
(1) For the text from Figure 2 to Figure 6, the description of figure number is incorrect. For example, in page 7 line 301, Figure 3 should be Figure 2.
Answer: In the original WORD manuscript, the figure numbers are all correct. However, in some unknown reason, the figures are incorrectly numbered in the PDF version. In the revised version, we will check the numbers in the PDF version.
(2) In the definition of SM1, Iij should be explained.
Answer: We add the definition of I_ij in line 537 of the revised manuscript.
(3) The equations of SM1, SM2 and SM3 should be numbered.
Answer: We add the equation numbers of SM1, SM2, and SM3.
(4) There is no conjunction between sentences.
Page 18 Line 655, There is still overlap errors in some objects, it is caused ...
Answer: The above sentence is rewritten as follows:
As shown in the figure, there is still overlap errors in some objects, which is caused by shape mismatch between the real object and the CAD model.
(5) Page 13 Line 559, is to finding
Answer: We correct it as: is to finding -> is finding
(6) Page 14 Line 576, next, step
Answer: We correct it as: the next step

Reviewer 3 Report
Add reference -> The first technique is to utilize the 3D depth information of the object in an image using a depth camera.
Add reference -> The second technique is to perform 3D-to-3D matching between the CAD model and the 3D shape information of the object extracted...
Both are maybe given the latter, but I am not sure.
The authors give as one of the main contributions that their pose pipeline uses only 4-DoF (reduction from 9-DoF). But, for comparison, the authors do not write how many DoF have already published works in 2. Literature Review uses? Something can be concluded from the titles of the references, but not for all.
References 38 and 39 are incomplete. Remove them, or add the required information.
The calculation time from the input RGB image to the output annotated image can be interesting, so authors should give tables with minimum, average, and maximum calculation time for various input RGB images.
It would also be interesting if the times were given for various steps in the calculation pipeline.
The authors should give a comment on why 10 objects (Bed, Motobike, Table, Oven, Bag, Clock, Bottle, Chair, Car, and Bench) are selected for comparison with reference [30]. The experimental setup from [30] uses 395 different models. Maybe their method is only better because fitting CAD models are selected.
Author Response
Dear Reviewers,
Thank you for your valuable comments on our manuscript, ”Relative Pose Estimation between Image Object and ShapeNet CAD Model for Automatic 4-DoF Annotation” submitted to ‘Applied Science’ journal. We have revised the manuscript according to all reviewer’s comments. In below, more revision details are shown in each comment from the reviewers.
Reviewer #3 :
Comments and Suggestions for Authors
Add reference -> The first technique is to utilize the 3D depth information of the object in an image using a depth camera.
Add reference -> The second technique is to perform 3D-to-3D matching between the CAD model and the 3D shape information of the object extracted...
Both are maybe given the latter, but I am not sure.
Answer: Most 3D pose estimation methods use the depth or 3D shape information of the object because the depth information provides strong clues to pose estimation. Thus, as above, we explain that depth cameras or stereo cameras are commonly used to obtain the depth information. In the recent review papers, many common approaches of 3D pose estimation are presented. Thus, in the end of the above sentences, we add three recent review papers as references as follows:
- He, Z.; Feng, W.; Zhao, X.; Lv, Y. 6D Pose Estimation of Objects: Recent Technologies and Challenges. Applied Sciences (Switzerland) 2020, 11.
- Gorschlüter, F.; Rojtberg, P.; Pöllabauer, T. A Survey of 6D Object Detection Based on 3D Models for Industrial Applications. J Imaging 2022, 8, doi:10.3390/jimaging8030053.
- Wang, Y.; Wang, C.; Long, P.; Gu, Y.; Li, W. Recent Advances in 3D Object Detection Based on RGB-D: A Survey. Displays 2021, 70, doi:10.1016/j.displa.2021.102077.
The authors give as one of the main contributions that their pose pipeline uses only 4-DoF (reduction from 9-DoF). But, for comparison, the authors do not write how many DoF have already published works in 2. Literature Review uses? Something can be concluded from the titles of the references, but not for all.
Answer: Not all literatures explain the exact DoF of their pose estimation methods. Most methods using a single RGB image address the 9-DoF estimation problem because the unknown scale between the CAD model and the image object. Other approaches using RGB-D images mostly address 6-DoF because of the scale of the image object is measured and known. Even though, this assumption is not always correct. In Section 2, we try to add DoF of some conventional methods, which have been described in their papers. Scan2CAD[13] is 9-DoF, Mask2CAD[14] is 5-DoF, Pix3D[15] is 9-DoF, and DenseFusion[22] is 6-DoF. Revised parts are marked in red-colored texts.
References 38 and 39 are incomplete. Remove them, or add the required information.
Answer: We add the URL and access information of the references.
The calculation time from the input RGB image to the output annotated image can be interesting, so authors should give tables with minimum, average, and maximum calculation time for various input RGB images.
It would also be interesting if the times were given for various steps in the calculation pipeline.
Answer: As described in Abstract and Introduction, the main purpose of the proposed method is building the 3D pose annotation database between image objects and their corresponding CAD models. Thus, it is not necessary the proposed method runs in real or fast time. This is the reason why there is no run-time analysis in the manuscript. In addition, as shown in Figure 1, the proposed method consists of several sub-steps (sub-tasks) and the data flow is complex. Thus, we run each sub-step one by one, by using the results from the previous step. Thus, in our current implementation, run-time analysis is not possible.
The authors should give a comment on why 10 objects (Bed, Motobike, Table, Oven, Bag, Clock, Bottle, Chair, Car, and Bench) are selected for comparison with reference [30]. The experimental setup from [30] uses 395 different models. Maybe their method is only better because fitting CAD models are selected.
Answer: In the reference [31], (originally [30]) total 395 difference models are used, however their CAD models belong to only 9 categories. (chair, sofa, table, bed, desk, bookcase, wardrobe, tool and miscellaneous). Thus, the category number is similar with ours. In addition, in our comparison, the triplet loss learning network in [31] has been trained by using the same number of training images, 592 RGB images, which is the almost same with our DELF-based similarity measurement. In Section 4.4, training and test of the triplet loss learning network has been described in detail.
